# Novel Physiology and Definition of Poor Ovarian Response; Clinical Recommendations

**DOI:** 10.3390/ijms21062110

**Published:** 2020-03-19

**Authors:** Antoine Abu-Musa, Thor Haahr, Peter Humaidan

**Affiliations:** 1Division of Reproductive Endocrinology and Infertility, Department of Obstetrics and Gynecology, American University of Beirut Medical Center, Beirut 1107 2020, Lebanon; 2The Fertility Clinic Skive Regional Hospital, 7800 Skive, Denmark; THOHAA@rm.dk (T.H.); peter.humaidan@midt.rm.dk (P.H.); 3Faculty of Health, Aarhus University, 8000 Aarhus C, Denmark

**Keywords:** ovarian stimulation, IVF, Poseidon criteria, Bologna criteria, poor ovarian response

## Abstract

Poor ovarian response (POR) to controlled ovarian stimulation (OS) presents a major challenge in assisted reproduction. The Bologna criteria represented the first serious attempt to set clear criteria for the definition of POR. However, the Bologna criteria were questioned because of the persistent heterogeneity among POR patients and the inability to provide management strategies. Based on these facts, a more recent classification, the POSEIDON (Patient-Oriented Strategies Encompassing IndividualizeD Oocyte Number) classification, was developed to provide a homogeneous and refined definition of POR that significantly reduces the heterogeneity of the Bologna criteria definition of POR and helps in the clinical handling and counseling of patients. In this review, we discuss the impact of the POSEIDON classification on the clinical management of patients with POR.

## 1. Introduction

The ovarian response to ovarian stimulation (OS) reflected by the number of oocytes retrieved is a keystone in in vitro fertilization (IVF) cycles and an independent factor in the success of treatment [1,2]. Although the ideal number of oocytes needed might be a matter of debate, 10–15 follicles is considered to be the optimal response after OS [1]. Patients with poor ovarian response (POR) present a major challenge in IVF, as they are not likely to obtain an optimal number of oocytes after OS, and thus are less likely to conceive and, moreover, will have a higher risk of cycle cancellation. The estimated prevalence of POR ranges from 6% to 35% [3,4]. This wide range is caused by the initial lack of consensus when defining POR. A systematic review [5], including a total of 47 studies performed in POR patients, revealed that 41 different definitions of POR were used, with differences in the criteria used for the definition and the values used for each criterion. This lack of uniformity led to the development of the Bologna criteria established by the European Society of Human Reproduction and Embryology (ESHRE) in 2011, which was the first attempt to set clear definitions of the POR segment [6]. According to the Bologna criteria, POR is defined as a condition where at least two of the following criteria are present: (i) advanced maternal age (>40 years), (ii) a previous poor response, defined as ≤3 oocytes retrieved after a standard OS, (iii) or an abnormal ovarian reserve test (antral follicle count (AFC) < 5–7 follicles and/or anti-Mullerian hormone (AMH) < 1.1 ng/mL). These criteria have been accepted by the scientific community and are currently the most commonly used to identify patients with POR. Despite this, criticism has arisen because the POR population defined by the Bologna criteria remains heterogeneous, and importantly, the criteria do not take the impact of age on oocyte quality and therefore success rates into consideration [7,8]. In addition, the Bologna criteria do not provide any recommendations for clinical decision-making, counseling, or management of the POR patient.

In 2016, a new classification by a panel of fertility experts known as the POSEIDON (Patient-Oriented Strategies Encompassing IndividualizeD Oocyte Number) group [9] was introduced, providing a more detailed stratification of the POR patient to reduce the heterogeneity seen in the Bologna POR population and to promote individualized treatment in these patients. The POSEIDON classification divides patients into four subgroups based on a combination of factors: (i) age, (ii) ovarian reserve markers, and (iii) ovarian response (oocyte number retrieved) in a previous OS cycle if this information is available (Figure 1). In practical terms, the POSEIDON classification stratifies patients with POR into two main categories, namely the “unexpected POR” (Group 1 and 2) and the “expected POR” (Group 3 and 4). It is estimated that 47% of all patients undergoing Assisted Reproductive Technology (ART) treatment belong to POSEIDON groups 1 to 4, thus accounting for a substantial subset of women attending ART clinics [10]. The primary goal of ART is to help infertile couples achieve a singleton live birth in the shortest time possible. One way to achieve this goal is to transfer a single euploid embryo at the blastocyst stage [11,12,13]. In this regard, the POSEIDON classification proposed a new marker of success, namely the ability to obtain the number of oocytes needed in each patient to obtain at least one euploid blastocyst for transfer [14]. The aim of the present review is to provide an overview of the POSEIDON classification and its role in suggesting treatment strategies in POR.

## 2. Measure of Success According to POSEIDON Concept

The success of ART treatment is increasingly being measured by the live birth rate (LBR) per transfer or per OS cycle, the so-called cumulative live birth rate (CLBR). As mentioned, The POSEIDON Group introduced a new measure of success, which is the number of oocytes needed to achieve at least one euploid blastocyst for transfer in the individual patient [14]. The availability of a euploid blastocyst for transfer may change the fate of the low-prognosis POR patient, as 50–60% of euploid blastocysts implant across all age categories [15]. Although it is true in general that patients producing more eggs [10,11,12,13,14,15] have better outcomes [1], it is not clear whether this applies to patients with POR, since randomized controlled studies (RCTs) are lacking. However, it was suggested that the retrieval of one more oocyte increases the predicted LBR per cycle [1]; having three oocytes instead of two leads to a relative increase in LBR by 25% in all age groups and even in patients with poor responses (<5 oocytes) and suboptimal responses (5–9 oocytes) [1,2]. The concept of “the more the better” and “one more oocyte matters” becomes particularly valid in patients with POR as the higher the number of oocytes collected/accumulated, the higher the chance of obtaining an embryo cohort that might include at least one euploid blastocyst [16,17]. Therefore, it is helpful to estimate the minimum number of oocytes needed per patient to achieve at least one euploid blastocyst. Two suggested tools that might be useful in assessing and increasing the probability of having one euploid blastocyst are the follicle-to-oocyte index (FOI) and the ART calculator.

### 2.1. Follicle-to-Oocyte Index (FOI)

The prediction of an ovarian response to OS is crucial for the optimal and individualized management of patients undergoing ART. Both AFC and AMH have been successfully used to predict both poor and hyper responses [18]). In contrast, the FOI is a new model introduced to address the hyporesponsiveness or ovarian resistance to gonadotropins during OS and or insufficient ovulation trigger strategies; see Figure 2 [19,20]. Basically, FOI assesses the ratio between the total number of oocytes retrieved at oocyte pickup to the number of antral follicles available at the beginning of stimulation (Figure 2). Thus, a low FOI is defined as less than 50% of antral follicles resulting in oocytes after OS. A low FOI can be seen in patients with adequate ovarian reserve markers, where only a fraction of the available antral follicles was recruited during OS. Therefore, patients with a history of a low FOI typically will benefit from another stimulation and/or ovulation trigger strategy in a subsequent OS with the potential to increase the recruitment of follicles [21], thus increasing the chances of achieving at least one euploid blastocyst for transfer.

### 2.2. The ART Calculator

As a pretreatment predictive model, the ART calculator (Figure 3) has been developed by the POSEIDON Group to assist estimating the number of oocytes needed to have one euploid blastocyst for transfer [22]. The ART calculator provides two types of predictions; one is used as a pretreatment estimation of the number of mature oocytes needed to achieve at least one euploid blastocyst, and another uses the actual number of collected mature oocytes to estimate the chances of having a euploid blastocyst using that oocyte cohort. This helps clinicians counsel patients about their prognosis and establish individualized treatment regimens. The ART calculator provides its estimation based on a number of predictors, mainly female age and type of sperm, which were found to be the most important factors for blastocyst euploidy. For further information regarding the ART calculator, see http://www.members.groupposeidon.com/Calculator.

## 3. POSEIDON Groups 1 and 2

POSEIDON groups 1 and 2 include women who had a poor (<4) or suboptimal (4–9) number of oocytes retrieved after OS despite the presence of an adequate ovarian reserve defined as AFC > 5 and/or AMH > 1.2 ng/mL. These patients are classified as group 1 if younger than 35 years or group 2 if ≥ 35 years of age. In practical terms, POSEIDON groups 1 and 2 are hyporesponders who unexpectedly show a low ovarian response or resistance to OS with exogenous gonadotropins. Both groups are likely to have a lower CLBR than normal or high responder patients [2]. In these patients the FOI is low, meaning that the number of oocytes retrieved at the end of stimulation is not consistent with the AFC available at the beginning of stimulation. The prevalence of this condition is not clear; however, it is estimated that around 10% of women believed to be normal responders based on ovarian reserve testing will require a higher dosing of gonadotropins or luteinizing hormone (LH) activity supplementation to achieve an adequate follicular growth [23]. Approximately 47% of patients attending ART clinics will fit the POSEIDON criteria [10], and it was previously estimated that of these 5% and 35% will belong to groups 1 and 2, respectively [10].

The pathophysiological mechanism of the hyporesponse noted in POSEIDON groups 1 and 2 is not fully understood; however, it seems that the decreased sensitivity of follicles to exogenous follicle-stimulating hormone (FSH) may be the most plausible explanation. In line with this some specific genetic polymorphisms, affecting gonadotropins and their receptors have been suggested. Thus, a common LH beta subunit variant was associated with increased FSH consumption during stimulation [24,25,26,27,28], and a systematic review and meta-analysis including 33 studies was recently published on the clinical relevance of FSH receptor (FSHR) polymorphisms and the correlation to response to treatment [28]. The analysis showed that both Serine carriers of FSHR (rs6166) and carriers of FSHR (rs1394205) are associated with ovarian resistance to stimulation.

Environmental contaminants might also affect ovarian response during OS [29,30,31]. Thus, Alviggi et al. in a retrospective study showed that elevated intrafollicular levels of benzene were associated with a decreased number of oocytes retrieved and embryos available for transfer [29], and it is currently believed that benzene leads to transduction deficiency in the FSHR. In another retrospective study, polychlorinated biphenyl congeners (PBC) in follicular fluid was associated with a decreased ovarian response during OS [32]. Finally, there is some evidence that oxidative stress might also affect both folliculogenesis and spermatogenesis [33,34]. Thus, oxidative stress leads to an increase in free radicals such as reactive oxygen species (ROS), which seem to influence the quality of oocytes, spermatozoa, and embryos [35] and therefore will negatively affect the ART outcome [33,36].

### 3.1. Clinical Management of POSEIDON Groups 1 and 2 Patients

Since the new marker of success in ART according to the POSEIDON group is to retrieve the number of oocytes needed to have at least one euploid blastocyst for transfer, the goal of the management of POSEIDON groups 1 and 2 is to find a way to increase oocyte yield and subsequently the reproductive outcome. According to the available literature, five main strategies should be considered for use alone or in combination, namely: (i) use of recombinant FSH (rFSH) in preference over urinary gonadotropins, (ii) an FSH dosage increase, (iii) the use of recombinant LH (rLH), (iv) dehydroepiandrosterone (DHEA) supplementation before OS, and (v) a double-stimulation (DuoStim) protocol. The suggested management of POSEIDON groups 1 and 2 is illustrated in Figure 4.

### 3.2. Use of rFSH

Multiple RCTs and meta-analyses have shown that the use rFSH is associated with significantly more oocytes retrieved as compared to urinary FSH formulations, both in long gonadotropin releasing hormone (GnRH) agonist (GnRHa) and GnRH antagonist cycles [37,38,39]. This is related to the higher biopotency of recombinant formulations [40]. Therefore, stimulation using recombinant formulations (rFSH) should be considered in all POSEIDON group 1 and 2 patients.

### 3.3. FSH Dose Increase

Increasing the FSH dose in patients with a history of suboptimal response is a commonly used strategy. This strategy might be used to rescue an ongoing OS in women with initial slow response to gonadotropins or in a subsequent stimulation cycle in women with a history of suboptimal response. In a retrospective study by Drakopoulos et al., a total of 160 women with a normal ovarian reserve and a history of suboptimal response (4–9 oocytes retrieved) were recruited to receive an increase in rFSH in the subsequent fixed GnRH antagonist co-treated cycle [41]. A significantly higher number of oocytes (9 versus 6, *p* < 0.001) and good quality embryos (4 versus 3, *p* < 0.001) was retrieved compared to the previous cycle. According to the authors, an increase of 50 IU of the initial rFSH dose would lead to the retrieval of one more oocyte. Even patients with FSHR polymorphism seem to benefit from increases in rFSH dosage. Thus, Behre et al. randomized Ser680/Ser680 carriers to receive a daily rFSH of either 150 IU or 225 IU. The 225 IU/day dose was able to restore estradiol levels of Ser680/Ser680 carriers similar to those of women with the wild-type genotype at the end of stimulation [42]. In conclusion, rFSH dose increases are effective in POSEIDON 1 and 2 patients.

### 3.4. rLH Supplementation

Several studies evaluated the addition of rLH to the OS protocol in women with ovarian hyporesponse [43,44,45,46,47,48]. Adding rLH as of days 7–10 to rescue an ongoing “slow” stimulation cycle might be more efficient than increasing the dosage of rFSH. In this line, De Placido et al. [45] in an RCT included 260 women undergoing OS following a long GnRHa downregulation protocol. With a starting dose of 225 IU rFSH, 130 patients showed signs of a slow response, which was defined as serum estradiol levels < 180 ng/mL and follicles < 10 mm in diameter on day 8 of stimulation. On this day, patients were randomized to receive either 150 IU rLH in addition to rFSH or to have an increase in the rFSH dose of another 150 IU. The number of oocytes retrieved was significantly higher in patients who received rLH supplementation (9.0 ± 4.3) compared those patients having an increase in their rFSH dosage (6.1 ± 2.6, *p* < 0.01). In addition, the implantation rate (14.2% versus 18.1%, *p* > 0.05) and ongoing pregnancy rates (32.5% versus 40.2%, *p* > 0.05) were similar to those observed in the control group, consisting of normal responders. These results were recently corroborated in an RCT by Yilmaz et al. [49], in which hyporesponders to OS were identified using the same criteria as in De Placido et al. [45]. Patients were randomized to receive either supplementation with 75 IU rLH or an increase of 75 IU in the rFSH dose. Pregnancy rates were significantly higher in the rLH supplementation (57.8%) and the control (64.7%) groups as compared to the increased dose rFSH group (32.4%, *p* < 0.02).

As for the rLH dosage, 150 IU rLH was previously proven to be superior to 75 IU rLH when a long GnRHa downregulation protocol was used [50]. Thus, in that RCT, hyporesponders similar to those reported by De Placido et al. [45] were randomized to receive either 150 IU or 75 IU of rLH, respectively. Patients receiving the 150 IU/day rLH had a significantly higher number of oocytes retrieved than those who received the 75 IU/day (9.65 ± 2.16 versus 6.39 ± 1.53, *p* < 0.05) [50]. However, it should be noted that the beneficial effect of adding rLH to OS until now was shown in studies using a long GnRHa downregulation protocol, and there is still no robust data on the use of rLH in GnRH antagonist cycles in patients with hyporesponses to rFSH only. The mechanism by which the addition of rLH improves ovarian response in patients with POR is not clear. The excessive suppression of endogenous LH after downregulation with a GnRH analogue is a plausible explanation, while another is related to the presence of polymorphisms in the LH molecule (LH β chain variant), reducing the bioactivity of the molecule or polymorphisms of the LH receptor. In line with this, Alviggi et al. [23,51] reported that carriers of the LH β chain variant had an ovarian resistance to exogenous gonadotropins and required higher dosages of rFSH during OS [23]. Importantly, this polymorphism of the LH molecule seems to be rather common worldwide with a carrier frequency ranging from 0 to 52%, depending on ethnicity.

The most recent systematic review and a further meta-analysis have shown that adding rLH to the OS protocol is beneficial for two subgroups of patients, (i) women with an adequate ovarian reserve having an unexpected hyporesponse to rFSH monotherapy, i.e., POSEIDON groups 1 and 2, and (ii) patients 36–39 years of age [47,52]. Therefore, adding rLH to OS in POSEIDON groups 1 and 2 patients should be considered, starting with 75–150 IU rLH either on days 7–10 of stimulation in an attempt to rescue an ongoing hyporesponse or from day one of stimulation in a subsequent cycle to increase the number of growing follicles—and subsequently the number of oocytes.

### 3.5. Dehydroepiandrosterone Supplementation

Pretreatment with DHEA before OS has been shown to significantly improve the birth rate in POR and patients of advanced age, according to the latest Cochrane meta-analysis [53]. As a precursor of testosterone, DHEA supplementation is believed to counterbalance the impaired theca function and androgen production observed in women of advanced age [54]. Tartagni et al. [55] conducted an RCT in 109 women fulfilling the POSEIDON group 2 criteria. Patients were randomized to receive DHEA supplementation 75 mg/day or placebo for a total of eight weeks before starting OS. Patients receiving DHEA had a significantly higher live birth rate compared to those who did not receive DHEA (22/53 versus 18/53, *p* < 0.05). In another RCT, Moawad et al. [56] reported a similar beneficial effect in patients receiving DHEA 75 mg/day for 12 weeks prior to OS compared to no treatment. Thus, the ongoing pregnancy rate was significantly higher (11/58 versus 7/47, *p* < 0.05) in patients who received DHEA pretreatment.

### 3.6. Double Stimulation

Double stimulation or so-called DuoStim combines conventional follicular phase stimulation with luteal phase stimulation during the same cycle. The aim of DuoStim is to maximize the number of oocytes obtained per cycle. A large multicenter study has shown comparable fertilization, blastocyst, euploidy, and pregnancy rates from follicular phase stimulation and luteal phase stimulation [57]. Moreover, the rate of patients obtaining at least on euploid blastocyst significantly increased from 42.3% after follicular phase stimulation only to 65.5% with DuoStim [57]. However, it should be noted that results from studies performed using the DuoStim protocol should be interpreted with caution, since most patients included in these studies do not explicitly fulfill the criteria of POSEIDON groups 1 and 2 but rather POSEIDON groups 3 and 4. However, from a clinical point of view, DuoStim might be a promising strategy even for POSEIDON group 1 and 2 patients who had an unexpectedly poor response to stimulation during their follicular phase stimulation.

### 3.7. Conclusions: Clinical Management POSEIDON Groups 1 and 2

POSEIDON groups 1 and 2 respond unexpectedly poorly to OS despite a good ovarian reserve. The main difference between these two groups of patients is their age and consequently a difference in oocyte euploidy and thus LBR. Group 1 is considered to have a better prognosis due to an age-related lower oocyte aneuploidy rate. Patients in group 2, although older and with an age-related higher oocyte aneuploidy, still have a good chance of reaching the estimated number of oocytes needed for one euploid blastocyst if properly stimulated. For both groups, this might be achieved by increasing the dosage of rFSH, rLH supplementation, and/or using a DuoStim protocol.

### 3.8. Future Research

The subdivision achieved in the POSEIDON classification can help us better understand the specifics of each homogeneous subgroup of patients undergoing ART. Future clinical trials should study each group separately aiming to find the optimal clinical management for each group. This will lead to the development of personalized treatments and tailored stimulation protocols. Research should evaluate the benefit of screening patients for specific FSH and LH receptor polymorphisms as well as the presence of variant LH-β to explore specific treatments for each trait. The role of practical indices such as FOI and the ART calculator in the management of POSEIDON groups 1 and 2 should also be further evaluated. Finally, research on the use of antioxidants to alleviate the possible negative effects of ROS on ovarian resistance to OS is warranted.

## 4. POSEIDON Groups 3 and 4

POSEIDON Groups 3 and 4 are patients who are expected to have a poor response to OS due to their low ovarian reserve in terms of a total AFC < 5, and/or AMH < 1.2 ng/mL. Group 3 includes patients < 35 years of age and group 4 ≥ 35 years of age. The latter group has a high risk of having poor oocyte quality in addition to a reduced quantity due to the age-related increase in oocyte aneuploidy [17]. Interestingly, the prevalence of POSEIDON groups 3 and 4 within the Bologna criteria population is 24% and 76%, respectively [58]. Among all POSEIDON groups, group 3 has been estimated to constitute 10% of patients and group 4 to constitute 55% of patients [10]. As mentioned, POSEIDON group 4 patients have a significantly poorer prognosis due to the age-related increase in oocyte euploidy, resulting in aneuploid embryos, higher embryo transfer cancellation rates, and the need for multiple IVF cycles. The management of these patients is difficult; however, the goal of treatment is to increase the probability of having at least one euploid blastocyst to transfer in the individual patient. This might be achieved by (i) choosing the appropriate individualized stimulation protocol, (ii) individualizing the ovulation trigger strategy, and (iii) although more evidence is needed, possibly adding adjuvant treatment to the OS protocol or prior to the initiation of OS. A summary of management strategies for groups 3 and 4 is shown in Figure 5.

### 4.1. Stimulation Protocols

Previously, it was suggested that natural cycle IVF or mild stimulation would be more optimal for the Poseidon group 3 and 4 patients when compared to conventional OS [59,60], as ovarian stimulation with high-dose exogenous gonadotropins was thought to increase the aneuploidy rates of oocytes and embryos. However, abundant recent scientific evidence does not support this concern [61,62,63,64]. Very low live birth rates per cycle of 2.6% have been reported in Bologna POR patients undergoing natural cycle IVF [65,66]. In contrast, an 11% live birth rate per cycle was reported in the until now largest RCT including Bologna POR patients, using a GnRH agonist long protocol and 300 IU rFSH in combination with 150 IU rLH daily from day one of stimulation [67]. Furthermore, recently, a 20% ongoing pregnancy rate was reported in poor ovarian reserve patients using follicular as well as luteal phase stimulation (DuoStim) [57,58]. To conclude, ovarian stimulation with exogenous gonadotropin stimulation at a maximum of 300 rFSH should be used rather than natural cycle IVF in POSEIDON groups 3 and 4 patients.

As for the choice of the stimulation protocol, Sunkara et al. reported a non-significant increase in the number of mature oocytes by one oocyte, and a significant decrease in cancellation rates when a long GnRH agonist protocol was used in patients with POR compared to a GnRH antagonist protocol [68]. This increase in oocyte yield is most probably caused by follicular synchronization obtained after downregulation, which is crucial for POSEIDON groups 3 and 4 patients, as they usually have increased late luteal phase FSH, promoting early recruitment of the leading follicle, and subsequently the suppression of those other few follicles left in the ovaries. Importantly, one more oocyte increases the live birth rate by 5% [1,69], making the long GnRH agonist protocol the preferred stimulation for expected POR. However, the inhibitory effect of late luteal phase endogenous FSH and thus early recruitment can also be achieved in GnRH antagonist cycles by a 5-day treatment with estradiol 4 mg, daily prior to menses, or oral contraceptives for 12–16 days, followed by a 5-day wash-out period [70,71]. When pretreatment with estrogen or oral contraceptives was compared to a long GnRH agonist protocol in two RCTs, there was no difference in the duration of stimulation, FSH consumption, number of embryos obtained, and pregnancy rates [70,71]. Moreover, double stimulation (DuoStim) might be considered to accumulate the number of blastocysts needed to increase the probability of having at least one euploid blastocyst for subsequent frozen thaw embryo transfer. Vaiarelli et al. [57] reported that a single DuoStim cycle in poor prognosis patients, essentially POSEIDON group 4 patients, resulted in 65.5% of patients having at least one euploid blastocyst for transfer [57]. Moreover, there was no difference in the ongoing pregnancy rate per single euploid blastocyst transfer between blastocysts obtained from the follicular phase and those obtained from the luteal phase stimulation, which were 39.5% and 49.4%, respectively [57]. In POSEIDON group 4 patients, an ongoing pregnancy rate per DuoStim of 20.7% can be considered highly successful in this challenging group [57].

Recombinant FSH is the preferred choice of gonadotropins in POR patients, and there is ample high-quality evidence showing that rFSH is superior to urinary FSH and human menopausal gonadotropin (hMG) as a means to increase the oocyte yield [37,38,39,72,73]. This is crucial, since the POSEIDON criteria of success relies on oocyte numbers to increase the chances of having at least one euploid blastocyst for transfer. As mentioned previously, the recommended maximum daily dose of rFSH is 300 IU, as higher doses do not increase neither the clinical pregnancy rate nor the live birth rate [74,75]. Corifollitropin alfa, a long-acting rFSH has the advantage of a rapid increase in FSH serum level, which leads to early recruitment and an increase in the number of pre-ovulatory follicles [58,76]. In an RCT, including Bologna POR patients, Corifollitropin alfa did not increase the pregnancy rate compared to rFSH; however, significantly more embryos were available for freezing [77], which is likely to increase the CLBR by increasing the chance of having at least one euploid embryo for transfer.

Finally, two systematic reviews suggested that rLH supplementation during OS is beneficial in women with hyporesponse to exogenous FSH-only stimulation and specifically in women aged 36–39 years [47,52]. Humaidan et al. [67] published the largest RCT in poor prognosis patients aligned with the POSEIDON group 4 patients. A total of 939 patients were randomized to either a fixed daily dose of 300 IU rFSH and 150 IU rLH or rFSH 300 IU alone. There was no significant difference in LBR between the two groups [67]. However, a post hoc analysis, stratifying patients into mild, moderate, and severe POR, showed that moderate and severe POR patients had a significantly higher LBR and a lower pregnancy loss rate when 150 IU rLH was added to the stimulation protocol from day one of stimulation [67]. Thus, physiological as well as clinical evidence suggests a beneficial effect of rLH supplementation in the POSEIDON group 4 patients.

### 4.2. Ovulation Trigger

In addition to the individualized OS, an individualized ovulation trigger (OT) strategy may also be used to increase the mature oocyte yield. One of the reasons for a low FOI could be mutations of the LH receptor [19,20,78], which might be improved by certain OT strategies. Final oocyte maturation and ovulation trigger is most commonly achieved using human chorionic gonadotropin (hCG). More recently, GnRH agonist trigger has been used in GnRH antagonist cycles to prevent ovarian hyperstimulation syndrome (OHSS) in OHSS risk patients. The LBR using GnRHa-triggered cycles is comparable to hCG-triggered cycles provided that a modified luteal phase support is applied [79,80,81]. Importantly, it was shown that the number of MII oocytes and embryos were significantly higher after a GnRHa trigger as compared to an hCG trigger [82,83,84]. The finding was supported by a recent systematic review and meta-analysis in which two RCTs showed a significant increase in the number of good quality embryos after a GnRHa trigger compared to an hCG trigger (MD 0.94, 95% CI 0.01–1.87) [85]. This finding probably is caused by the fact that a GnRHa trigger elicits a surge of endogenous LH as well as FSH, adding further physiological benefit for the activation of oocyte maturation [86].

Two new OT strategies, a dual trigger and double trigger, have been introduced for the management of a low FOI or a low mature follicle/mature oocyte rate. These two strategies have been suggested to improve IVF outcomes by overcoming impairment in follicular function, oocyte meiotic maturation, and cumulus expansion [20]. In dual triggers, both GnRHa and low-dose hCG are administered simultaneously [87], while in double triggers, GnRHa and hCG are administered at 40 and 34 h, respectively, prior to oocyte pick up [88]. Both strategies combine the advantages of GnRHa and hCG in terms of LH activity and early phase luteal support mediated by hCG and the endogenous secretion of LH and FSH mediated by GnRHa [89]. In addition, a double trigger might increase the maturity rate of retrieved oocytes in some patients who need a longer interval between the OT and oocyte retrieval to allow cumulus expansion and detachment of the oocyte [90].

### 4.3. Adjuvants

#### 4.3.1. Androgens

Pretreatment with androgens could be considered in patients with expected POR. The biological evidence has shown that androgens induce FSH receptors on granulosa cells [91], which subsequently leads to an increase in the recruitability and growth of pre-antral and antral follicles through Insulin-like Growth Factor 1 (IGF-1) [92,93]. According to the latest Cochrane meta-analysis, moderate quality evidence shows that DHEA and testosterone pretreatment may improve LBR in POR patients [53]. However, the dosages and timing of pretreatment have not yet been defined. An international clinical trial TTRANSPORT TRIAL for Bologna POR patients (clinicaltrial.gov identifier NCT02418572) currently evaluates a daily dose of 5.5 mg transdermal testosterone as a pretreatment for 60 days, and the results of this trial might provide further evidence for the possible clinical use of androgens in POSEIDON group 3 and 4 patients.

#### 4.3.2. Growth Hormone

The rationale for the use of growth hormone (GH) in ART is the fact that GH itself and through IGF-1 has a synergistic effect with FSH on follicular development [94,95]. Adjuvant treatment with GH in patients with POR has been shown to increase the number of oocytes retrieved and decrease gonadotropin consumption [96,97,98]. A Cochrane review in 2010, including four small studies with a total of 165 non-Bologna POR patients, reported a significant increase in LBR (OR 5.39, 95% CI, 1.89–15.35) in favor of GH compared to standard treatment [96]. Moreover, a more recent meta-analysis including 12 studies showed a significant increase in the number of oocytes retrieved (OR 1.94, 95% CI, 1.19–2.69) with GH supplementation compared to controls, whereas no increase in LBR was seen (OR 1.54, 95% CI, 0.86–2.74) [99]. On the other hand, a RCT by Norman et al. included women with poor response who failed to produce >5 oocytes in previous IVF cycles with >250 IU/day of FSH [100]. Patients were randomized to receive 12 IU of GH or placebo from the first day of stimulation. There was no difference in live birth rate between the GH group (9/62, 14.5%) or placebo (7/51, 13.7%) (OR 1.07, 95% CI 0.37–3.10). Therefore, due to the equivocal results and the relatively few and small RCTs conducted until now, the need for further large RCTs regarding the use of GH in POSEIDON groups 3 and 4 patients is underlined.

#### 4.3.3. Coenzyme Q 10

The hypothesis behind using coenzyme Q 10 (CoQ 10) is the reduction in mitochondrial oxidative stress improving oocyte competence. A recent RCT assessed the effectiveness of CoQ 10 pretreatment for 60 days prior to OS in POSEIDON group 3 patients [101], reporting a significant increase in the number of oocytes retrieved in the CoQ 10 supplemented group (4 (mean), interquartile range (IQR) 2–5) compared to controls (2 (mean), IQR 1–2). In addition, the CoQ 10 group had more high-quality day 3 embryos assessed by cytoplasmic fragmentation and size of blastomeres. Despite the fact that CoQ 10 and other antioxidants have almost no adverse reactions and side effects [101], more studies are definitely needed to better evaluate their use as pretreatment in POSEIDON groups 3 and 4 patients.

### 4.4. Conclusions: Clinical Management POSEIDON Groups 3 and 4

Due to the limited ovarian reserve, POSEIDON group 3 and 4 patients have a low reproductive prognosis and pose a clinical challenge. The current best practice to manage these patients is to individualize OS in order to increase the oocyte yield, which is essential to optimize the LBR. Even increasing the oocyte yield from 2 to 3 results in a 25% relative increase in the LBR across all age groups. Ongoing pregnancy rates above 20% per cycle of DuoStim in these difficult groups could be considered a successful achievement; however, more research is needed to suggest firm clinical recommendations.

### 4.5. Future Research

Randomized controlled trials (RCTs) taking into consideration the novel patient stratification suggested by the POSEIDON Group are warranted. Thus, future trials investigating the most optimal OS strategy for groups 3 and 4 are needed. This might be achieved through studies comparing the long GnRH agonist downregulation protocol to estrogen or contraceptive pill primed GnRH antagonist protocols and the DuoStim protocol. The outcome of these studies should include the cumulative LBR and the ability to obtain the number of oocytes needed in each induced patient to obtain at least one euploid blastocyst for transfer. More studies are also needed in the area of pretreatment with adjuvants, including androgens, GH, and antioxidants that explore the timing, duration, and dosages of the treatment. On a more experimental basis, the future management of the expected POR might include in vitro follicle activation [102], autologous mitochondrial transfer to improve the implantation rates and quality of embryos [103], and the development of oocytes from stem cells [104].

## 5. Conclusions

The POR patient constitutes a major challenge in reproductive medicine. The recently proposed POSEIDON criteria for POR stratifies patients into four homogeneous groups, leading to a better understanding of the different subgroups of POR, better development of individualized treatment regimens, and better research design. Groups 1 (<35 years) and 2 (≥35 years) are the unexpected POR patients with a good ovarian reserve, and groups 3 (<35 years) and 4 (≥35 years) are those who are expected to have POR due to a low ovarian reserve. The primary goal of the management of the POSEIDON POR patient is to maximize the oocyte yield so as to increase the likelihood of having at least one euploid blastocyst for transfer. This demands an individualized approach at all steps of the ART treatment, including a targeted pretreatment evaluation, the choice of GnRH analogues, gonadotropin type, and doses, ovulation trigger, and the possible use of adjuvant therapies. Tools such as FOI and the ART calculator can help develop individualized treatment regimens and most importantly help counsel patients about their reproductive prognosis. With this approach, even the very poor prognosis patients in POSEIDON group 4 may achieve an ongoing pregnancy rate of more than 20% per cycle (DuoStim). Future well-designed RCTs are warranted to evaluate the stratifications and the recommendations set forward in this review. In this aspect, POSEIDON groups 1 and 2 need to be studied separately from POSEIDON groups 3 and 4.

## Figures and Tables

**Figure 1 ijms-21-02110-f001:**
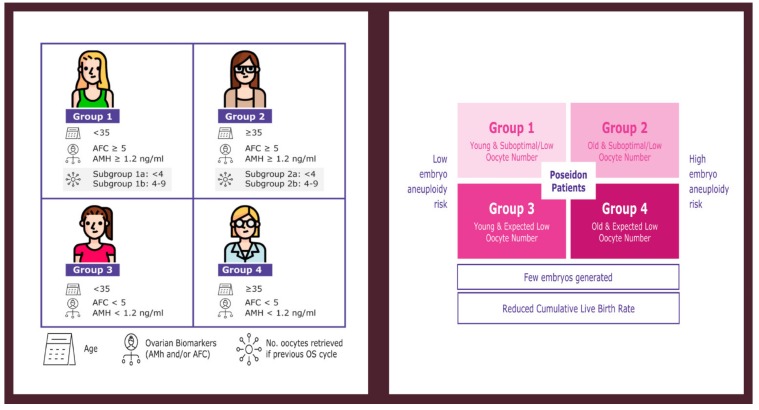
Patient-Oriented Strategies Encompassing IndividualizeD Oocyte Number (POSEIDON) criteria of low prognosis patients in Assisted Reproductive Technology (ART). The novel system relies on female age, ovarian reserve markers, ovarian sensitivity to exogenous gonadotropin, and the number of oocytes retrieved, which will both identify the patients with low prognosis and stratify such patients into one of four groups of women with “expected” or “unexpected” impaired ovarian response to appropriate exogenous gonadotropin stimulation. According to these criteria, four distinct groups of low prognosis patients can be established (left). Owing to low oocyte numbers and less embryos being produced, POSEIDON patients have lower cumulative live birth rates per started cycle than non-POSEIDON counterparts. However, the prognosis is differentially affected by female age as it relates to the risk of embryo aneuploidy (right). Art drawing by Chloé Xilinas. Modified from Esteves et al. Front. Endocrinol. Esteves, S.C.; Roque, M.; Bedoschi, G.M.; Conforti, A.; Humaidan, P.; Alviggi, C. Defining low prognosis patients undergoing assisted reproductive technology: POSEIDON criteria—the why. *Front. Endocrinol. (Lausanne)*
**2018**, *9*, 461. doi:10.3389/fendo.2018.00461. This is an open-access article distributed under the terms of the Creative Commons Attribution License (CC BY).

**Figure 2 ijms-21-02110-f002:**
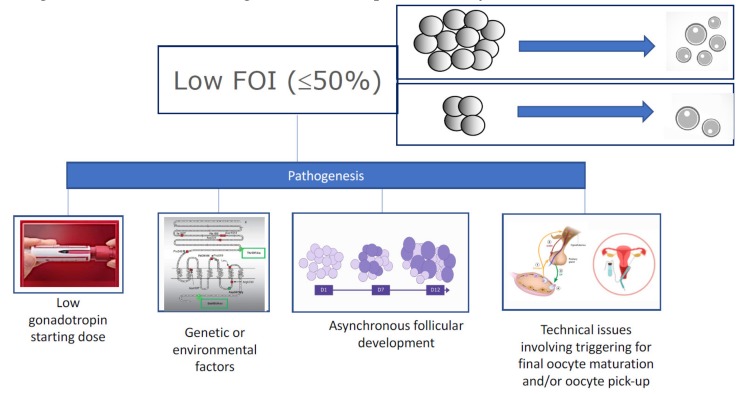
Pathogenesis of low follicle-to-oocyte index (FOI). Reprint from: Alviggi, C.; Conforti, A.; Esteves, S.C.; Vallone, R.; Venturella, R.; Staiano, S.; Castaldo, E.; Andersen, C.Y.; De Placido, G. Understanding Ovarian Hyporesponse to Exogenous Gonadotropin in Ovarian Stimulation and Its New Proposed Marker—The Follicle-To-Oocyte (FOI) Index. *Front. Endocrinol. (Lausanne)*
**2018**, *9*, 589. This is an open-access article distributed under the terms of the Creative Commons Attribution License (CC BY).

**Figure 3 ijms-21-02110-f003:**
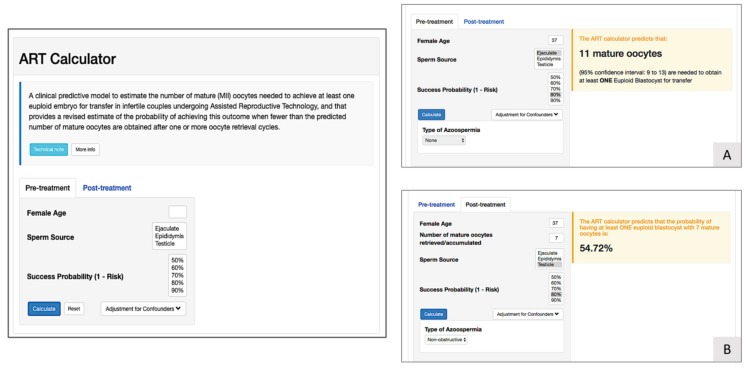
ART Calculator to estimate the minimum number of mature oocytes required to obtain at least one euploid blastocyst for transfer in infertile patients undergoing in vitro fertilization/intracytoplasmic sperm injection (IVF/ICSI) cycles. (**A**) Pretreatment, clinicians should input the patient age and the sperm source to be used for IVF/ICSI. If the option ‘Testicle’ is marked, then the type of azoospermia should be defined also. The probability of success is set by the user and indicates the chance of having ≥1 euploid blastocyst when the predicted number of mature oocytes is achieved. Its complement is the risk—that is, the chance of having no (zero) euploid blastocysts when the predicted number of oocytes is achieved. Once the button ‘Calculate’ is pressed, a text box will pop up on the right side of the screen, indicating the predicted minimum number of mature oocytes needed for obtaining at least one euploid blastocyst, with its 95% confidence interval. (**B**) Posttreatment, i.e., when fewer than the predicted number of mature oocytes are obtained after one or more oocyte retrieval cycles, clinicians should input the pretreatment information and the actual number of mature oocytes collected or accumulated. As in the pretreatment model, the probability of success is set by the user. Once the button ‘Calculate’ is pressed, a text box will pop up on the right side of the screen, indicating the predicted probability of achieving ≥1 euploid blastocyst with the number of mature oocytes available. Reprint from: Esteves, S.C.; Carvalho, J.C.; Bento, F.C.; Santos, J. A novel predictive model to estimate the number of mature oocytes required for obtaining at least one euploid blastocyst for transfer in couples undergoing in vitro fertilization/intracytoplasmic sperm injection: The ART Calculator. *Front. Endocrinol.*
**2019**, *10*, 99, doi:10.3389/fendo.2019.00099. This is an open-access article distributed under the terms of the Creative Commons Attribution License (CC BY). The online ART calculator can be found at http://www.members.groupposeidon.com/Calculator/.

**Figure 4 ijms-21-02110-f004:**
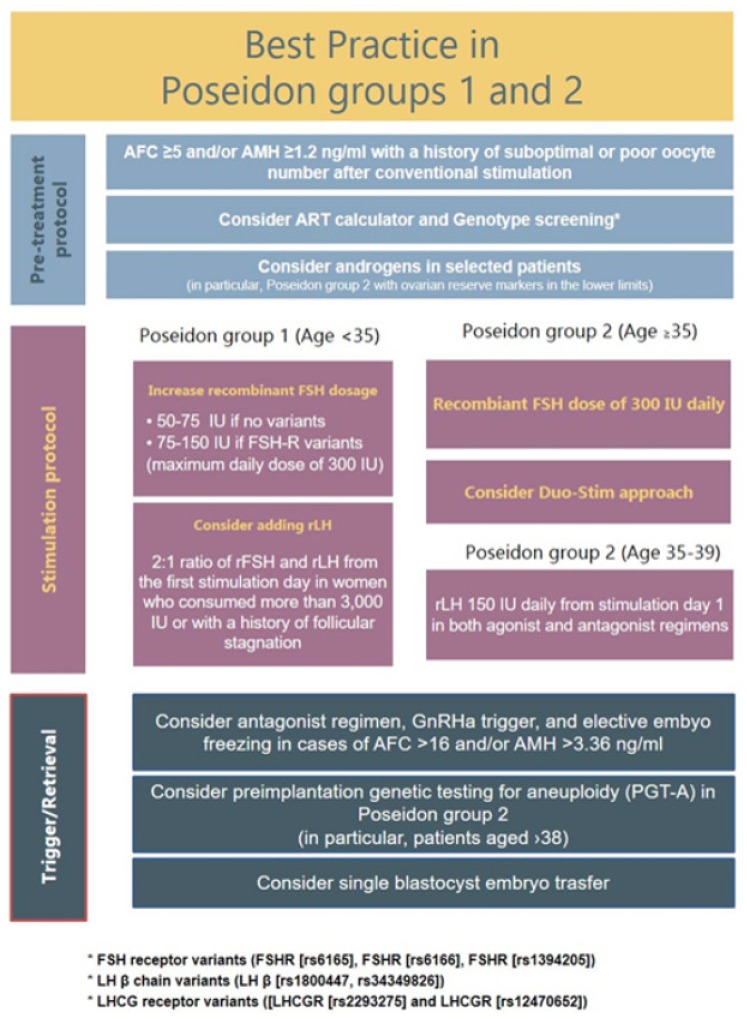
Best practice in POSEIDON groups 1 and 2. POSEIDON recommends the use of gonadotropin releasing hormone (GnRH) antagonist co-treatment for all patients in POSEIDON groups 1 or 2. Ovarian stimulation strategy: First choice should be an increase in the daily rFSH dose by 75–150 IU. In POSEIDON group 1, patients with follicular stagnation at follicle sizes 1–12 mm, rLH 75–150 IU daily should be added from day 1 of stimulation. In POSEIDON group 2, rLH 75–150 IU daily should be added to all patients from day 1 of stimulation. Ovulation trigger strategy: GnRH agonist (GnRHa) is mandatory in the follicular phase stimulation of the DuoStim protocol. All trigger agents can be used in the luteal phase stimulation. In non-DuoStim GnRH antagonist cycles, the choice of trigger between GnRHa and human chorionic gonadotropin (hCG) should rely on the embryo transfer strategy (fresh or frozen), patient characteristics (e.g., hypo–hypo) and clinical experience. In cases with a low FOI, clinicians should consider pretreatment including short-term estrogen, progestin therapy, or oral contraceptive pills (OCP) for synchronization of the follicles prior to stimulation, adjuvant LH activity during stimulation, or changing the trigger strategy to either dual or double trigger. In case of an insufficient number of oocytes retrieved as determined by the ART calculator, the probability of transferring a euploid embryo should be discussed with the patient to counsel whether an immediate transfer or a new stimulation should be suggested. Reprint from: Conforti, A.; Esteves, S.C.; Cimadomo, D.; Vaiarelli, A.; Di Rella, F.; Ubaldi, F.M.; Zullo. F.; De Placido, G.; Alviggi, C. Management of women with an unexpected low ovarian response to gonadotropin. *Front. Endocrinol. (Lausanne)*
**2019**, *10*, 387. doi:10.3389/fendo.2019.00387.

**Figure 5 ijms-21-02110-f005:**
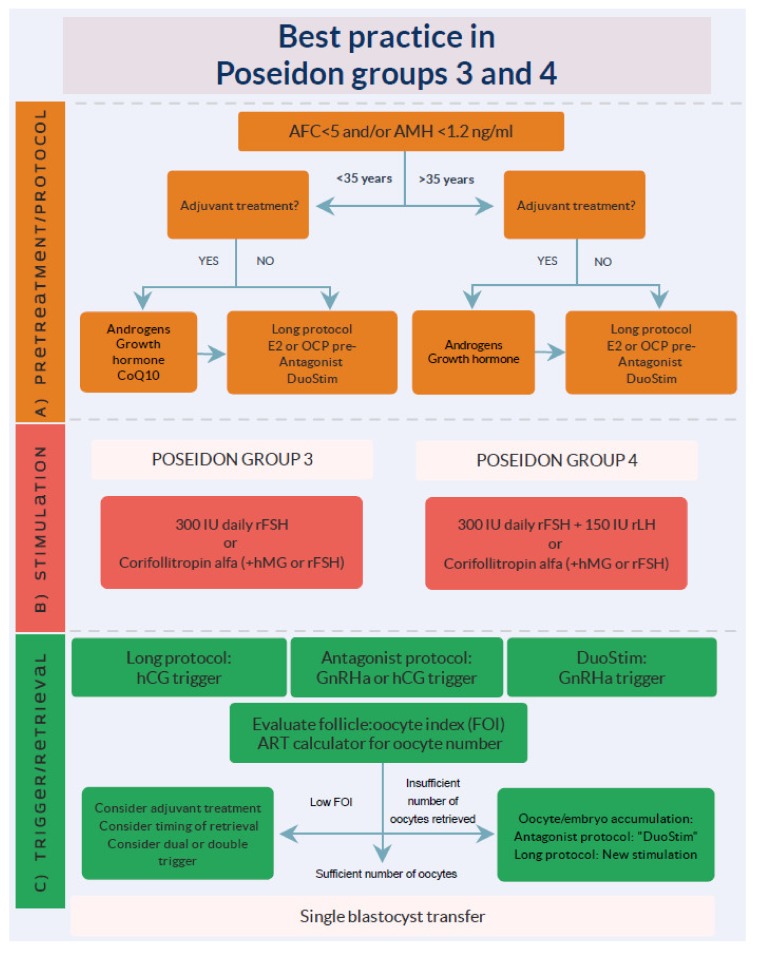
Management of POSEIDON groups 3 and 4. A) Pretreatment is rarely the first option in low-prognosis patients, but in case of inadequate ovarian response to stimulation, pretreatment should be considered. The choice should rely on the availability, clinical experience, and patient preference. Stimulation protocol might start using GnRH antagonist co-treatment, keeping in mind the possibility of converting to DuoStim to increase the chances of achieving the individualized number estimated by the ART calculator. Otherwise, a long GnRHa protocol should be considered as the first choice. B) Ovarian stimulation strategy: The first choice in Poseidon group 3 is the GnRH antagonist cycle with either 300 IU daily of recombinant FSH (rFSH) alone or combined with either recombinant LH (rLH) or human menopausal gonadotropin (hMG). In POSEIDON group 4 patients, 300 IU rFSH combined with 75–150 IU rLH daily should be given from day one of stimulation unless the combination of Corifollitropin alfa and hMG was chosen. The antagonist cycle allows the use of Duostim, unlike the long-agonist GnRH analogue. C) Ovulation trigger strategy: In the long GnRHa downregulation protocol, hCG is mandatory as an ovulation trigger. In Duostim, the GnRHa trigger is mandatory in the follicular phase stimulation, but both GnRHa or hCG trigger, or a dual trigger, can be used as an ovulation trigger in the luteal phase stimulation. In non-DuoStim GnRH antagonist cycles, the choice of trigger between GnRHa and hCG should rely on the embryo transfer strategy (fresh or frozen), patient characteristics (e.g., hypo–hypo), and clinical experience. In cases with a low FOI, clinicians should consider pretreatment including short-term estrogen therapy, synthetic progesterone, or OCP for synchronization of the follicles prior to stimulation, adjuvant LH activity during stimulation, or changing the trigger strategy to either dual or double trigger. Reprint from: Haahr, T.; Dosouto, C.; Alviggi, C.; Esteves, S.C.; Humaidan, P. Management Strategies for POSEIDON Groups 3 and 4. *Front. Endocrinol.*
**2019**, *11*, 99. doi:10.3389/fendo.2019.00614. This is an open-access article distributed under the terms of the Creative Commons Attribution License (CC BY).

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
