# Peer review of "Novel Physiology and Definition of Poor Ovarian Response; Clinical Recommendations"

_ijms, 2020, doi:10.3390/ijms21062110_

Round 1

Reviewer 1 Report

The concept of "The Poseidon Criteria" including their way to treat the poor responders in IVF

is relatively new. Although on a clinical base I agree with the four criteria of classification of poor responders

I'm  skeptic regarding  the clinical efficiency of the treatment approach.

The use of estrogen, progesterone or combined oral contraceptives as a pretreatment of these patients,

although logical from a theoretic point of view is not efficient and even wasteful when come to treatment:

It requires more gonadotropins ( compensatory) and decreases the efficiency of the GnRH-agonist as a trigger

( m/p due to a pituitary effect as measured in lower serum LH).   

I'm not aware of any prospective randomized study showing that accumulating eggs/embryos with subsequent

PGT-A is more efficient or money saving when compared with PGT-A done after each separate cycle.

The last but not the least is the statement that more eggs ( in these patients) means better results.

Again I don't know any prospective randomized study who can prove it. Is true in general that patients producing

more eggs ( up to 12-15) have better IVF results; is it true for these specific patients?

increasing the gonadotropins dosage with subsequent increased number of oocytes will result automatically in

higher pregnancy rates?

I'm afraid that we are confusing essentials with cosmetics.

In conclusion', I believe that this new approach is patient supportive but not patient effective.

In any case the authors must show the contrary in well designed prospective studies.

So far from my point of view the review is based on speculations mainly.  

Author Response

Reviewer 1 raised 3 main points:

1. The use of estrogen, progesterone or combined oral contraceptives as a pretreatment of these patients, although logical from a theoretic point of view is not efficient and even wasteful when come to treatment: It requires more gonadotropins (compensatory) and decreases the efficiency of the GnRH-agonist as a trigger (m/p due to a pituitary effect as measured in lower serum LH).   

The use pretreatment with either estrogen or combined oral contraceptives is a short duration treatment and is used to help in follicular synchronization. The two reference articles (72,73) were RCT that have shown that there is no difference in IVf outcome when pretreatment with estrogen and ocp was compared to long GnRH agonist cycles. This was mentioned in the text page 11 line 369 to read as follows: “ When pretreatment with estrogen or oral contraceptives was compared to long GnRH agonist protocol in two RCTs there was no difference in the duration of stimulation, FSH consumption, number of embryos obtained and pregnancy rates (72,73).”

2. I'm not aware of any prospective randomized study showing that accumulating eggs/embryos with subsequent PGT-A is more efficient or money saving when compared with PGT-A done after each separate cycle.The last but not the least is the statement that more eggs (in these patients) means better results. Again I don't know any prospective randomized study who can prove it. Is true in general that patients producing more eggs ( up to 12-15) have better IVF results; is it true for these specific patients? increasing the gonadotropins dosage with subsequent increased number of oocytes will result automatically in higher pregnancy rates?                                                                                                                 We fully agree with the reviewer that there are no RCT to prove that more  eggs/embryos accumulated in poor responders will increase the pregnancy rate. This was stated on page 3 line 80 as follows: “Although it is is true in general that patients producing more eggs (10-15) have better outcome (1), it is not clear whether this applies to patients with POR since RCTs are lacking.”

3. In conclusion', I believe that this new approach is patient supportive but not patient effective. In any case the authors must show the contrary in well designed prospective studies.

Agree that more RCTs are needed to evaluate the suggested management of Poseidon groups. This was clearly mentioned in the section Future Research on pages 9 and 13. In addition, it was further emphasized by changing the last sentence in the Conclusions section page 14 line 495 to read as follows: “Future well designed RCTs are warranted to evaluate the stratifications and the recommendations set forth in this review. In this aspect, POSEIDON groups 1 and 2 need to be studies separately from groups 3 and 4.”

Reviewer 2 Report

The review by Abu-Musa et al. is a well written work summarizing the key research on the field. The review is comprehensive and well constructed. No major concerns exist, however, some statements throughout lack references. Please check for the presence of appropriate references throughout the review.

Author Response

The review by Abu-Musa et al. is a well written work summarizing the key research on the field. The review is comprehensive and well constructed. No major concerns exist, however, some statements throughout lack references. Please check for the presence of appropriate references throughout the review.

The manuscript was reviewed few statements needed backup references already included in the study.